# Social and Ecological Dimensions of Urban Conservation Grasslands and Their Management through Prescribed Burning and Woody Vegetation Removal

**Alison Farrar [1], Dave Kendal [2,]* [ID], Kathryn J. H. Williams [3] and Ben J. Zeeman [4]**

[1]  Department of Environment, Land, Water and Planning, Warrnambool 3280, Australia;
   alison.farrar@delwp.vic.gov.au
[2]  School of Technology, Environments and Design, University of Tasmania, Hobart 7000, Australia
[3]  School of Ecosystem and Forest Sciences, The University of Melbourne, Melbourne 3010, Australia;
   kjhw@unimelb.edu.au
[4]  Glenelg Hopkins Catchment Management Authority, Hamilton 3300, Australia; b.zeeman@ghcma.vic.gov.au
*  Correspondence: dave.kendal@utas.edu.au

**Abstract:** Natural grasslands are threatened globally. In south-eastern Australia, remnants of critically endangered natural grasslands are increasingly being isolated in urban areas. Urbanisation has led to reduced fire frequency and woody plant encroachment in some patches. Grasslands are currently being managed under the assumption that desirable management actions to address these threats (prescribed burning and removing woody vegetation) (1) lead to improved conservation outcomes and (2) are restricted by negative public attitudes. In this study, we tested these two assumptions in the context of native grassland conservation reserves in Melbourne, Australia. Firstly, we investigated differences in species and functional trait composition between patches that had been recently burnt, patches that were unburnt and patches subject to woody vegetation encroachment. We found that the functional traits of species converged in areas subject to woody plant encroachment and areas frequently disturbed by fire. Burning promoted native species, and patches of woody plants supressed the dominant grass, providing a wider range of habitat conditions. Secondly, we surveyed 477 residents living adjacent to these grassland conservation reserves to measure values, beliefs and attitudes and the acceptance of prescribed burning and removing woody vegetation. We found conflict in people's attitudes to grasslands, with both strongly positive and strongly negative attitudes expressed. The majority of residents found prescribed burning an acceptable management practice (contrary to expectations) and removing trees and shrubs from grasslands to be unacceptable. Both cognitive factors (values and beliefs) and landscape features were important in influencing these opinions. This research provides some guidance for managing urban grassland reserves as a social–ecological system, showing that ecological management, community education and engagement and landscape design features can be integrated to influence social and ecological outcomes.

**Keywords:** threatened ecological community; biodiversity; species richness; native plants; environmental values; environmental beliefs; environmental attitudes; urban conservation reserves

## 1. Introduction

Temperate grasslands are among the world's most human-altered biomes and have been identified as the biome most in need of conservation attention globally [1]. Urbanisation is a major threat to grasslands worldwide [1,2] and transforms grasslands through processes such as habitat loss, changes

in land-use and the removal of natural disturbance regimes [3,4]. In Australia, a key focus of grassland conservation is the preservation of endemic species and ecological communities which can be out competed by introduced species, particularly in conjunction with altered disturbance regimes. As a consequence of urban expansion, some important native grassland remnants are now confined to isolated patches in or around cities [2]. This proximity to human populations influences management decision making, particularly about management actions perceived to be controversial, such as biomass reduction through burning and tree removal.

## 1.1. Fire in Grasslands

Fire is an important natural disturbance event in temperate grasslands. Without frequent fire, dominant grasses will competitively exclude inter-tussock species, reducing biodiversity [5,6]. Maintenance of an open sward is required to prevent the competitive exclusion of inter-tussock species for water and light resources and to preserve a high level of plant species richness [5,7]. This is particularly important in Southern Australia, as many inter-tussock grassland plants do not possess a persistent seed bank, and thus, loss of species from the standing vegetation will not be naturally reversed with the reintroduction of disturbance [5,8]. Studies have shown that a frequent fire regime (every 1–3 years) facilitates species coexistence, maintains or increases native plant diversity and reduces exotic species diversity, particularly in highly productive grasslands [7,9].

As human activities transform landscapes, there has been a global trend of reduced fire frequency in temperate grasslands [2,10]. This trend is particularly stark for grassland remnants confined to cities and towns, as prescribed burning close to residential areas is perceived to be difficult and fires that occur are suppressed [2,11]. In urbanised grasslands, natural processes have also been replaced with alternative management regimes such as livestock grazing or slashing, further changing grassland community structure and composition [12]. This decline in fire frequency has, in part, resulted in significant changes in grassland community structure and composition [13].

The consequences of returning fire to long unburnt ecosystems have to date produced both positive and negative outcomes for grasslands in Southern Australia. For example, Sinclair et al. [14] recorded significant tussock mortality when examining a 'hot' summer fire in long unburnt modified C3 native grassland, while another study found native grassland species were not killed when re-introducing fire under 'cooler' weather conditions and where C4 grasses were dominant (Zeeman unpublished data). Factors such as drought, timing of fire and species identity are likely to be central in determining grassland resilience to a reintroduced burning regime [15].

In order to achieve the reintroduction of fire into a landscape, it is commonly agreed that social constraints on prescribed burning must be understood [16,17]. However, relatively few studies have explored social perceptions of prescribed burning in grasslands [16,18–20]. To date, research has largely focused on perceptions of private landowners toward prescribed burning in grasslands on private property in agricultural landscapes [18–20]. These studies identify risk, and liability concerns tend to underpin negative perceptions of the practice [16,18–20]. However, few studies have explored public perceptions of prescribed burning in grasslands in an urban context. McGee found mixed and relatively low support for prescribed burning as a wildfire management tool in a large nature reserve consisting of grassland and forest ecosystems in Edmonton, Canada [21]. Similarly, studies in a forest context have also found variation in public perceptions of prescribed burning [22]. Some studies have found public favour toward prescribed burning in forests [23,24], while others found conflicted public support and scepticism toward the practice, particularly near urban areas [25–27].

## 1.2. Woody Plant Encroachment in Grasslands

The altered biotic and abiotic conditions transforming grasslands in urban areas have also been attributed to the increased density and cover of trees and shrubs, a global phenomenon known as woody plant encroachment. The effects of woody plant encroachment can be profound, reducing

species richness of ground cover plants, driving the loss of large widely spaced trees and altering fire behaviour [28–30].

Recent studies demonstrate woody and herbaceous plants can also interact via positive (facilitation) mechanisms [31–33]. Eldridge and Soliveres suggest that the effects of encroachment are scale-, species- and environment-dependent and that effects on ecosystem functioning are highly dependent on the species encroaching and density of encroachment [33]. It is possible that shading from woody plants supresses dominant grasses, providing habitat conditions suitable for more shade-tolerant species [34] (Peterson and Reich, 2008). Herbaceous species possessing the more efficient C4-photosynthetic pathway are likely to be disproportionally disadvantaged as light resources become increasingly limited under the canopy of trees, compared to species possessing a C3-photosynthetic pathway which tolerate lower light intensities and tend to be favoured in shady habitats [35].

Social constraints also exist on the removal of woody vegetation from grasslands. Sharp et al. suggest restoring vegetation communities modified by woody plant encroachment can be difficult as these altered communities can be valued in various and conflicting ways by different people and stakeholders [36]. For example, when restoring grasslands in an urban context, Alario suggests tree eradication programs may be a point of public conflict and may lead to outrage when "woody weeds" are removed [37,38]. This is not surprising given the various studies predicting landscape preferences suggest people generally hold treeless ecosystems, such as grasslands, in low regard [39].

### 1.3. Public Attitudes to Urban Grasslands and Their Management

There is increasing recognition that conservation outcomes in part depend on the way people think about the environment [40]. As endangered temperate grasslands increasingly share space with people, understanding the urban public's perceptions of grasslands and their ecologically important management actions is critical if grasslands are to be effectively conserved into the future.

Studies have shown that people's attitudes (i.e., positive or negative evaluations) to landscapes and acceptance of management (i.e., disposition toward a particular management action) are shaped by cognitive responses to landscapes [22,41–43]. A growing body of research describes the cognitive relationships between an individual's values (i.e., guiding principles about what is morally preferred), valued attributes of the landscape (i.e., properties of landscapes that are important to people such as biodiversity or cultural features) [44] and beliefs about the consequences of management actions (i.e., a subjective belief about what is true) regarding nature and the environment [41,45,46]. These concepts are commonly organised in a hierarchy with values being a more stable construct central to personality and are likely to influence beliefs which are more open to change, which in turn influence much more changeable attitudes [47].

While people's values and beliefs can influence their response to landscape management, many studies have identified designed features of landscapes (e.g., public access, interpretive signage and facilities) can also mediate attitude and acceptability judgements [48–50]. Nassauer suggests ecologically valuable native landscapes may not be protected or maintained if human intention to care for the landscape is not apparent [48]. Nassauer proposes landscapes designed with 'cues to care' offer a powerful vocabulary to communicate ecological function and quality, often referred to as 'landscape language' [48]. For example, Williams [38] suggests native grasslands with weedy edges, cyclone wire fences and no formal indication of management may lack 'cues to care' that help to illustrate the quality in these 'messy ecosystems'. Understanding the drivers of public attitudes can help managers to predict public response to management [51]. The extent to which public responses are cognitive or based on the physical features of the landscape has significant implications for management [50]; management should focus on landscape design approaches when public response is based on the physical features in the landscape and on management planning if responses are largely guided by people's values and beliefs. This study explores the extent to which public attitudes toward grasslands and their management are cognitive or based on the designed physical features of grasslands to help to inform grassland conservation in urban areas.

*1.4. Research Questions*

There are two important and related knowledge gaps in our social and ecological understanding of urban grasslands this study aims to address:

1.  What are the ecological effects of reintroducing fire and removing woody plants from urban grasslands? These management actions are assumed to be ecologically beneficial as they reverse processes known to degrade grasslands, yet the evidence to support decision making is mixed. At the same time, there are assumed to be social constraints on these management actions.
2.  What attitudes do the neighbouring public hold toward urban grasslands and the ecologically important actions of management burns and removing woody plants from urban grasslands, and what factors shape these attitudes? In particular, what is the relative importance of cognitive (values and beliefs) and landscape design features?

Anecdotally, effective grassland management is challenged by the general public's negative attitudes toward urban grasslands and toward management actions assumed to be beneficial, such as burning and woody plant removal. However, there is empirical evidence on the views of the public needed to support decision making.

## 2. Methods

*2.1. Conceptual Framework—Environmental Filtering*

The habitat filtering hypothesis suggests plant communities are a result of environmental filters (e.g., climate, soil) that constrain the functional type of species that are able to persist in a given habitat [52–54]. This theory predicts convergence in the functional traits of species based on the local abiotic environment [52,54]. By modifying habitat and altering disturbance regimes, the functional composition and diversity of plant communities may change as the abundance of species with traits maladapted to novel habitat conditions decline and better-adapted species increase [53]. Applying this theory to the context of highly productive temperate grassland in urban areas, we hypothesise that as consequence of the decline in fire frequency (creating a more competitive environment) [2], the frequency and abundance of species with traits with a competitive advantage (e.g., height, high specific leaf area (SLA)) will increase, whereas conditions within woody plant patches (creating a stressed environment) may favour stress-tolerant species possessing traits that enable plants to persist in resource-poor conditions (i.e., low SLA, slow-growing and long-lived), while filtering out fast-growing, competitive plants that employ resource-expensive strategies [55].

*2.2. Conceptual Framework—Cognitive Hierarchy*

A conceptual framework loosely based on Ford et al.'s [43,50] framework for public acceptability of management was developed to explain attitude and acceptability judgements of prescribed burning and removing woody vegetation from grasslands in urban areas. The framework distinguishes between two possible drivers of attitudes and acceptability judgements: (1) values and beliefs of individuals (cognitive processes) and (2) designed features of grasslands (responses to physical features in the landscape). These concepts were explored separately to identify the extent to which people's attitudes are driven by the landscape design of grasslands, which would have management implications for the design of grasslands in urban areas, compared to the extent to which attitudes are driven by the values and beliefs of individuals which may have more strategic and planning implications for management regarding public engagement addressing beliefs about grasslands and their management.

*2.3. Study Area*

The study was undertaken in temperate native grasslands in the northern and western suburbs of Melbourne, Victoria, Australia. These grasslands represent the eastern distribution of the Victorian Volcanic Plain (VVP), which stretches across Southwestern Victoria covering 2.3 million ha. At the

time of European settlement, the VVP was dominated by grassland. However, the ecosystem has been extensively cleared for agriculture. In recent decades, some of the largest remnants have come under pressure from Melbourne's expanding urbanisation. Currently, less than 2.1% of the pre-European extent remains intact, and the community is listed as critically endangered under Australian State and Federal legislation [2]. The grasslands were historically dominated by the C4 grass, *Themeda triandra*, with herbaceous species predominantly from the Asteraceae, Lilliaceae and Fabaceae families occupying inter-tussock spaces [56].

Melbourne's remaining native grasslands are generally small, fragmented patches that have been set aside for conservation. Historically, these grasslands were managed for different purposes (e.g., grazing, rural residential living, roadsides, cemeteries, military barracks), under different forms of ownership and under the influence of diverse stakeholders and legislative requirements [11]. The reserves also vary in landscape context, being surrounded variously by industrial, residential, commercial or peri-urban agricultural land and bordered by roads, freeways, railways or rubbish tips. Several reserves also share their setting with cemeteries. The reserves often differ in designed landscape features, with some containing a path (either inside or bordering the reserve), a fence (some with a 3 m high cyclone wire fence while others have no fence), or facilities (such as seats, playgrounds, barbeques or shelters, while others have no facilities).

## 3. Study 1—Ecological Effects of Reintroducing Fire and Woody Plant Removal

### 3.1. Site Selection

From the total pool of publicly managed grasslands in Melbourne (n = 80), sites were selected where (i) part of the site had been burnt less than one year ago ('recently burnt', n = 14) or (ii) the site contained a patch of woody vegetation (trees or large shrubs) ('woody plant halos', n = 9). In both cases, sites were only selected when they also contained a patch that had been burnt >3 years ago ('unburnt'), noting that the fire frequency in these grasslands was historically very high and fires were likely to occur every 1–3 years [57]. Areas within woody plant halos that had been burnt were not surveyed as there were not enough replicates to assess this treatment. Stoney knolls and areas within 5 m of the reserve boundary were not surveyed to reduce confounding influences from habitat conditions and edge effects. The selected study sites varied from less than 1 to 164 ha in size. Previous studies have shown that all reserves, including small ones, have high levels of endemic species diversity [58].

### 3.2. Field Surveys

Sites were surveyed from October to December 2014, when most vascular species were in flower to aid identification and detectability. At each study site, ten $1 \times 1$ m quadrats were randomly placed within each management treatment (recently burnt, unburnt and woody plant halos). The woody plant halo management treatment was defined as the area within a 10 m radius from the base of all woody plants (trees or large shrubs) within the patch [59]. All species detected in each quadrat were identified and recorded. The floristic composition of vascular plants was assessed in a total of 370 quadrats.

### 3.3. Trait Data Collection

Species trait values were collected for traits known to be related to environmental conditions: plant height (cm), seed mass (mg), specific leaf area (SLA: mg/mm$^2$), leaf dry weight (mg), photosynthetic pathway (C3/C4), life span (annual/biannual or perennial) and life form (a ten-point classification based on the Raunkiaer life form categories). Trait values were obtained from the Supplementary Information from Cross et al. [60] (SLA, leaf dry weight and height), from the TRY Plant Trait Database [61] (SLA, leaf dry weight, seed mass, photosynthetic pathways, life form and life span) and from field guides [62] (height, life span and life form). For life span and life form traits not available in the literature, local grassland experts [11] were consulted for their expert opinion of these trait values. Trait data for

plant height, seed mass, SLA and leaf dry weight were first averaged per treatment, per site, and these site values were used in analyses to avoid pseudo-replication.

### 3.4. Data Analysis

*Species diversity*: Simpson's Index of Diversity was calculated for the three management treatments: recently burnt, unburnt and areas within a patch of woody plant halos using Microsoft Excel macro software developed by Lepš et al [63]. The differences between management treatments were analysed using a one-way ANOVA and Bonferroni Post Hoc procedure (in the statistical package SPSS v22) based on the total species pool and native and exotic species separately.

*Species composition*: Species composition for the three management treatments was calculated and compared using the PRIMER software package [64,65]. Analysis of similarity (ANOSIM) was calculated to test for significant differences in species composition between management treatments. Similarity percentage (SIMPER) analysis was performed to determine the contribution of each species to the average dissimilarity between the management treatments. Species were only listed if they contributed to more than 50% of the cumulative rank order dissimilarity tally between the management treatments. Average percent similarity of species present in each management treatment was calculated using the Sørensen similarity index.

*Functional diversity*: To compare the functional diversity (the diversity of functional traits) of species in response to the three management treatments—unburnt (competitive environments), areas within woody plant halos (stressed environments) and recently burnt (disturbed environments)—community trait-weighted means were calculated using a modified Mason index [63] to compare the overall variation of trait values for plant height (cm), seed mass (mg), specific leaf area (SLA) (mg/mm$^2$) and leaf dry weight (mg). A modified Rao index of diversity [63] was also calculated to measure the dissimilarity of traits between species for plant height (cm), seed mass (mg), SLA (mg/mm$^2$), leaf dry weight (mg), life span (annual/biannual or perennial) and life form (a ten-point classification based on the Raunkiaer life form categories) [66]. The differences in the frequency of C3/C4 photosynthetic pathway of graminoids between management treatments were analysed using a one-way ANOVA and Bonferroni Post Hoc procedure based on the total species pool and native and exotic species separately to better understand the effects of the treatments on the threatened ecological community (only native species are protected).

## 4. Study 2—Social Attitudes toward Grasslands and Their Management

### 4.1. Survey Participants and Survey Distribution

All residential addresses within 100 m of a grassland conservation reserve in residential areas in Melbourne (n = 38) were selected from the Department of Environment, Land, Water and Planning (DELWP) property parcel database using ESRI ArcMap GIS software (version 10.2). The resident who was next to celebrate their birthday and who is over the age of 18 was invited to fill in and return the questionnaire (n = 2832). A modified version of Dillman's [67] approach for survey distribution was used, with participants receiving an initial mail out containing the survey material and a reply-paid envelope, and after two weeks, a follow-up reminder postcard was sent to non-respondents to increase the response rate. The use of real stamps was also used to increase response rates.

### 4.2. The Survey Instrument

The questionnaire used in this survey contained four sections:

1. *Values for natural areas in cities*: adapted from an existing scale, the Valued Attributes of Landscape Scale (VALS) [44] was used to measure how attributes of natural areas in cities are important to people. The general themes in this modified version of VALS included concepts of social values (e.g., accessibility, safety); natural values (e.g., providing habitat); cultural and heritage values (e.g., learning about cultural traditions, seeing historic things); experiential values (e.g., sights

and smell, relaxing atmosphere); productive use values (e.g., firewood collection, picking berries and mushrooms); and commercial use values (e.g., space for shops and houses). The VALS was adapted to an urban context by removing timber and food production items and adding new items related to commercial property development and recreation. Participants were asked to rate how important they thought these 22 attributes of natural areas in cities were to them on a 7-point Likert scale ranging from "Not important at all" to "Very important".

2.  *Beliefs about the consequences of management actions* to capture participants' beliefs about the outcomes of two ecologically important management actions in temperate grasslands: prescribed burning and allowing regrowth of woody vegetation (trees and shrubs) in grasslands. This was a new scale adapted from previous work on beliefs about consequences of management actions [36,68]. The same scale items were used for each management action. Concepts explored in this scale included positive consequences for grasslands (e.g., is needed for the long-term survival of the grassland), positive consequences for people (e.g., creates a good place for people to meet and socialise) and negative consequences for people (e.g., makes the landscape look uncared for). A 7-point Likert scale ranging from "Don't agree at all" to "Strongly agree" was used. Participants were also invited to make any open-ended comments on these management practices.

3.  *Acceptability of using prescribed burning and removing woody plants from grasslands*: to capture how acceptable the use of prescribed burning and removing trees and shrubs in urban grasslands was to respondents. Responses were measured on a 7-point Likert scale ranging from "Not acceptable" to "Very acceptable" and included an "I don't know" option.

4.  *Attitudes toward urban grasslands*: to capture the respondent's attitudes toward and awareness of the grassland in their neighbourhood. The attitudes scale included items on satisfaction, feelings of safety, enjoyment of use and maintenance measured on a 7-point scale ranging from "Don't agree at all" to "Strongly agree", with an "I don't know" response also included. Self-reported knowledge of grasslands was measured on a 7-point scale ranging from "Not knowledgeable at all" to "Very knowledgeable". Participants were also invited to make any open-ended comments regarding the grassland reserve in their neighbourhood.

Some basic demographic questions were also asked, such as gender, age, level of education, cultural background (indicated by whether they spoke a language other than English, a commonly used measure of cultural diversity in Australia [69]). These demographic questions allow comparison of the sample with the broader population. Two indicators of respondents' interest in the grasslands were also captured: whether they have been involved with community events for their local grassland and whether they have worked in the environmental sector (see Supplementary Materials for the final version of the survey).

### 4.3. Designed Landscape Attributes

Landscape design features are particularly important in grassland design as grasslands can be poorly appreciated and perceived as weedy, monotonous and hazardous [39,70]. Several landscape features were identified as likely predictors of people's attitudes toward grasslands, including style of fence considered a structural 'cue of care' indicating human attention to the landscape. If the fence is well designed and well maintained, the landscape is more likely to appear well cared for [11,48,70]; interpretive signs considered an important knowledge intervention aimed at engaging and teaching the public about important ecological features, drawing attention to forms of stewardship that may not be readily apparent in the landscape or encouraging people to accept ecological benefits as a fair trade for aesthetic shortcomings [49]; paths and facilities considered important because allowing access promotes engagement and preventing access promotes feelings of exclusion and enmity [70]. Publicly available data coding the landscape features of Melbourne's publicly managed grasslands were used [71] (Table 1).

**Table 1.** Scoring categories used to assign values for landscape features at the 23 sampled grassland reserves (derived from [71]).

| Landscape Design Feature | Score Categories |
| --- | --- |
| Fence | 1 = no fence, 2 = fence with open gate, 3 = short fence (0–1.2 m tall) with gate, 4 = high fence (above 1.2 m tall) with gate |
| Facilities | 1 = no facilities, 2 = minimal facilities such as external seat, rubbish bin, 3 = club or friends group facilities, 4 = notable public facilities such as play equipment, bbq facilities, shelters) |
| Information | 1 = no information, 2 = small sign stating conservation reserve, 3 = small education sign, 4 = large board or extensive interpretation |
| Paths | 1 = no paths, 2 = some external paths, 3 = major external paths, 4 = paths through grassland |

*4.4. Analysis*

Attitudes toward grasslands and acceptability of management actions were examined using frequency distributions, a simple but effective way of interpreting public views [72]. The shape of response frequency distributions was interpreted based on methods by Ribe [22]. A uni-modal distribution with one peak describes public consensus. A bi-modal distribution with two peaks describes conflicted public opinion. The overall level of acceptability is determined by the mean of the distribution.

Exploratory Factor Analysis (EFA) using Maximum Likelihood analysis (Varimax rotation) was conducted to validate the VALS in the context of natural areas in cities [44,73]. A separate EFA using Principal Components Analysis (Oblimin rotation with Kaiser Normalisation) was performed to identify groups of survey items that behaved similarly for people's beliefs about the consequences of prescribed burning and removing woody vegetation in grasslands. The number of components to extract was chosen by examining scree plots. Items were assigned to components when pattern matrix loadings were >0.40, and component scores were calculated as the mean of all items loading on each component. All data were analysed using IBM's SPSS 2013 version 22.

Relationships among the way people value natural areas in cities, beliefs about the consequences of using prescribed burning and allowing woody vegetation regrowth in grasslands, self-reported knowledge of the neighbourhood grassland, acceptance of management actions and attitudes toward the local grassland were determined via separate linear regressions on acceptability component scores and attitude item values. Value and belief component scores, knowledge of the local grassland item values and the landscape design features ordinal scores (i.e., fence, path, information and facilities) were included as independent variables.

## 5. Result

*5.1. Study 1—Ecological Effects of Reintroducing Fire and Woody Plant Removal*

A total of 104 vascular plant species were recorded across the sampled management treatments (62 native and 42 exotic). Total species diversity was slightly but significantly higher in burnt treatments compared to unburnt treatments ($F_{2,34} = 3.88$, $p = 0.03$; Figure 1a). There was no significant difference in native species diversity between the management treatments ($F_{2,34} = 1.74$, $p = 0.19$). Native species composition was significantly different between areas within woody vegetation halos and between areas that were recently burnt and unburnt areas (ANOSIM $p < 0.05$). Exotic species composition was significantly different between areas within woody vegetation halos and recently burnt areas (ANOSIM $p < 0.05$).

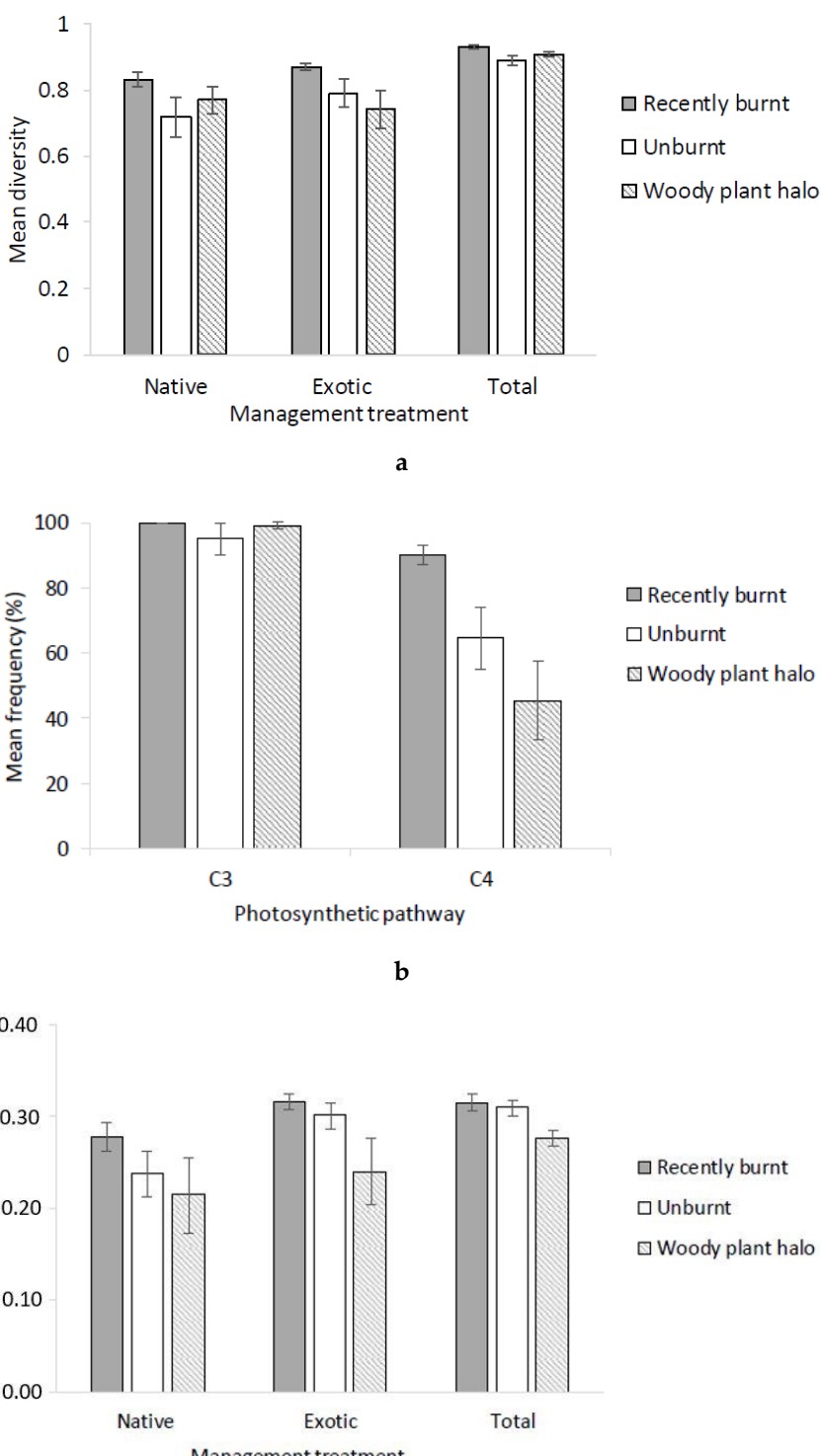

**Figure 1.** *Cont.*

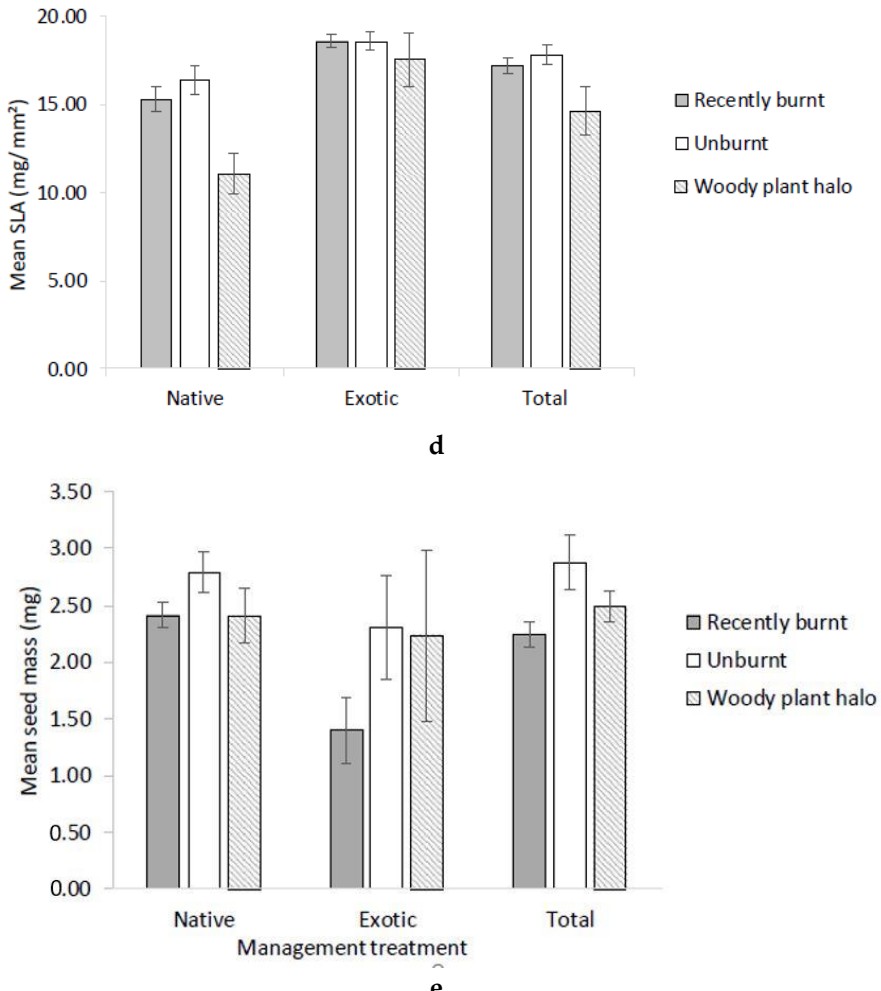

**Figure 1.** Comparison of recently burnt, unburnt and woody plant halos for (**a**) Simpson's Index of Diversity (mean diversity ± 1 SE, n = 104; (**b**) photosynthetic pathways of graminoids (mean ± 1 SD, n = 33; (**c**) Rao index of species height diversity (mean ± 1 SE, n = 104; and community trait-weighted means (n = 71) of (**d**) SLA (mg/ mm$^2$) (mean ± 1 SE; (**e**) seed mass (mg) (mean ± 1 SE).

Seventeen species, twelve of which were exotic, contributed up to 50% of dissimilarity between the management treatments (Table 2). The native graminoid, *Themeda triandra*, possesses a C4-photosynthetic pathway and has relatively high SLA (20.76 mg/ mm$^2$) compared with C3 graminoids and was more common in recently burnt management treatments compared to woody plant halos (Table 2), while the native graminoids, *Rytidosperma* spp. and *Austrostripa* spp., which possess C3 photosynthetic pathways and have low SLA (2.59 and 2.95 mg/ mm$^2$ respectively), were more common in woody plant halos (Table 2). Graminoids possessing a C4-photosynthetic pathway were significantly more frequent in recently burnt management treatments compared to woody plant halos (F2,35 = 7.01, *p* = 0.001) and unburnt treatments (F2,35 = 7.01, *p* = 0.02) (Figure 1b). There were no differences in the frequency of graminoids possessing a C3-photosynthetic pathway between the management treatments (F2,35 = 0.73, *p* = 0.49). Recently burnt treatments recorded significantly greater diversity in total species plant height compared to areas within woody plant halos (F2,34 = 4.23, *p* = 0.03; Figure 1c), as measured by the Rao index of trait diversity. This difference is driven by exotic species, with the diversity of plant height in exotic species in recently burnt areas being significantly more than that in exotic species within woody plant halos (F6,34 = 4.28, *p* = 0.02).

**Table 2.** Results of similarity percentage (SIMPER) analysis showing species frequency within woody plant halos, unburnt areas and recently burnt areas. The species contributing 50% of the dissimilarity between management treatments are shown. * Analysis of similarity (ANOSIM) *P* < 0.05. # Denotes exotic species.

| Species | Woody Plant Halo. | Unburnt. | Recently Burnt. |
|---|---|---|---|
| *Themeda triandra* | 44 * | 73 | 87 |
| *Plantago lanceolata* # | 37 * | 66 | 69 |
| *Rytidosperma spp* | 50 * | 10 * | 39 |
| *Romulea rosea* # | 59 | 51 | 92 * |
| *Briza maxima* # | 21 * | 37 | 40 |
| *Avena barbata* # | 26 | 36 * | 26 |
| *Austrostipa spp* | 34 * | 16 | 15 |
| *Briza minor* # | 20 * | 30 | 56 * |
| *Nassella neesiana* # | 22 | 16 | 19 |
| *Aira caryophyllea* # | 17 | 21 | 50 * |
| *Bromus hordeaceus* # | 9 | 21 | 14 |
| *Vulpia bromoides* # | 14 | 19 | 36 * |
| *Senecio quadridentatus* | 8 | 11 | 17 * |
| *Hypochaeris radicata* # | 0 * | 12 | 23 |
| *Sonchus oleraceus* # | 13 | 4 * | 19 |
| *Anthoxanthum ororatum* # | 1 * | 12 | 12 |
| *Veronica gracilis* | 3 | 6 | 17 * |

*5.2. Community Trait-Weighted Means*

Mean species SLA was significantly lower in woody halos compared to unburnt areas (F2,34 = 0.02; Figure 1d). Differences in SLA are driven by native species, with native species SLA being significantly lower in woody halo areas compared with unburnt areas (F2,34 = 9.47, *p* = 0.000; Figure 1d) and recently burnt areas F2,34 = 9.47, *p* = 0.01). Mean species seed mass in unburnt areas was significantly lower in recently burnt areas (F2,34 = 3.51, *p* = 0.04; Figure 1e). These were the only significant difference recorded between the management treatments for community trait-weighted means.

## 6. Study 2—Social Attitudes toward Grasslands and Their Management

There were 477 returned surveys (response rate of 17%). This response rate is consistent with other surveys of the public in the same region [74,75], allowing the relationship among values, beliefs about consequences, acceptability and attitude judgements, but not average opinions of the broader community. In comparison to the 2011 census data for Local Government Areas containing the grassland reserves (Brimbank, Hume, Darebin, Hobsons Bay, Melton, Moreland, Moonee Valley and Wyndham), the achieved sample was slightly more females (59% vs. 51%), slightly older (mean age category of 35–44 vs. mean 33) and had a similar proportion of respondents speaking a language other than English at home (43% vs. 44%).

*6.1. Factor Analysis of Valued Attributes for Natural Areas in Cities*

The factors in the VALS are reasonably well understood [44]; however, the changes made to the scale by incorporating new items to measure values related to urban development were expected to alter the factor structures. EFA was therefore used and revealed this to be the case, finding six groups of values (Table 3) explaining 61.0% of variation in the data: Four factors remained the same as the original

VALS: (1) natural values (e.g., threatened plants and animals, large old trees); (2) experiential values (e.g., peace and tranquillity, beauty); (3) cultural and heritage values (e.g., human history, culture); and (4) social values (e.g., visitors, accessibility). Two new factors were identified: (5) commercial use values (e.g., using the land for shops, developing the land for new housing); and (6) recreational setting values (e.g., mountain bike riding and horse riding). A reliability analysis revealed acceptable internal reliability for the factors with Cronbach's alpha ranging from 0.88 and 0.69 (Table 3).

**Table 3.** Loading values for valued attributes of landscape components. The survey items are phrased as they appeared in the survey instrument. Only loadings >0.40 are reported (n = 446).

| Survey Items | Cultural Heritage | Commercial Use | Natural | Social | Recreational Setting | Experiential |
|---|---|---|---|---|---|---|
| A place for human history and stories | 0.99 | | | | | |
| Learning about cultural traditions | 0.77 | | | | | |
| A place to see historic things such as old railways, rock art or stone walls | 0.48 | | | | | |
| Developing the land for new housing | | 0.95 | | | | |
| Using the land for shops, cafes and car parks | | 0.71 | | | | |
| Native plants and animals | | | 0.86 | | | |
| Large old trees | | | 0.73 | | | |
| Beautiful sights, sounds and smells | | | 0.68 | | | |
| Healthy land and waterways in which natural processes can continue | | | 0.50 | | | |
| Spaces for people to exercise, e.g.; jogging, walking | | | | −0.84 | | |
| A place for a short walk | | | | −0.67 | | |
| A place that is safe for people to visit | | | | −0.66 | | |
| Being accessible for people | | | | −0.60 | | |
| Spaces for people to interact and socialise | | | | −0.59 | | |
| Activities such as mountain biking or horse riding | | | | | 0.69 | |
| Motorised activities such as trail bike riding or four-wheel driving | | | | | 0.65 | |
| Activities such as fishing or collecting firewood | | | | | 0.47 | |
| Utilising the land for active recreation, e.g.; sports fields and courts, BMX tracks, skate parks | | | | | 0.45 | |
| Getting away from the stresses of everyday life | | | | | | 0.51 |
| Gathering food such as mushrooms, herbs and berries | | | | | | 0.44 |
| Experiencing nature through activities such as sight-seeing or bird watching | | | | | | 0.44 |
| **Reliability (Cronbach's Alpha)** | 0.86 | 0.82 | 0.88 | 0.84 | 0.73 | 0.69 |
| **Portion variance explained** | 29% | 11% | 9% | 5% | 4% | 3% |

*6.2. Factor Analysis for Beliefs about the Consequences of Management Actions*

The EFA of beliefs about consequences of using prescribed burning in grasslands in cities resulted in a three-component solution that explained 75.7% of the variation in the data (Table 4). EFA of beliefs about the consequences for allowing trees and shrubs to grow in grasslands resulted in a similar four-component solution that explained 78.3% of the variation in the data. The components are described as (1) having a positive benefit for people (prescribed burning and allowing trees and

shrubs to grow); (2) having negative consequences for people (prescribed burning or allowing trees and shrubs to grow); (3) having positive benefits for grasslands (prescribed burning); and (4) creating a good environment for people (allowing trees and shrubs to grow). A reliability analysis revealed acceptable internal reliability for the components with Cronbach's alpha values ranging from 0.84 to 0.70 (Table 4).

**Table 4.** Loading values for beliefs about consequences of (a) using prescribed burning in grasslands and (b) allowing trees and shrubs to grow in grasslands. Only loadings >0.40 are reported (n = 445).

| Survey Items | Positive Benefits for People | Negative Consequences for People | Positive Benefits for Grasslands | Positive Environment for People |
|---|---|---|---|---|
| **(a) Using prescribed burning in grasslands:** | | | | |
| Creates a good place for people to meet and socialise | 0.90 | | | |
| Creates a good place for people to exercise | 0.87 | | | |
| Is good for people health | 0.78 | | | |
| Is dangerous for people | | 0.89 | | |
| Makes the landscape look uncared for | | 0.85 | | |
| Is needed for the long term survival of the grassland | | | 0.91 | |
| Creates habitat for native animals, birds and insects | | | 0.82 | |
| Protects rare or threatened plants and animals living in the grassland | | | 0.62 | |
| Reliability (Cronbach's Alpha) | 0.84 | 0.70 | 0.79 | |
| Portion of variance explained (%) | 47% | 18% | 11% | |
| **(b) Allowing trees and shrubs to grow in grasslands:** | | | | |
| Is needed for the long term survival of the grassland | | | 0.93 | |
| Rejuvenates the grassland | | | 0.85 | |
| Protects rare or threatened plants and animals living in the grassland | | | 0.63 | |
| Makes the landscape look uncared for | | 0.91 | | |
| Is dangerous for people | | 0.87 | | |
| Creates a good place for people to exercise | 0.84 | | | |
| Creates a good place for people to meet and socialise | 0.80 | | | |
| Is good for people's health | | | | 0.86 |
| Creates a beautiful looking landscape | | | | 0.80 |
| Creates habitat for native animals, birds and insects | | | | 0.73 |
| Reliability (Cronbach's Alpha) | 0.92 | 0.74 | 0.82 | 0.85 |
| Portion of variance explained (%) | 10% | 17% | 44% | 8% |

*6.3. Attitudes toward Melbourne's Grasslands and Their Management*

The distribution analysis revealed neutral but conflicted public attitudes toward grasslands in Melbourne (Figure 2). Residents tended to have either strongly positive or strongly negative views about their local grassland. Positive attitudes toward neighbourhood grasslands were expressed through survey comments such as: "l love this area ... l have no excuses for [not] walking the dog every morning, walking in the fresh air before I begin work. Also having young children being able to show them the native grasses and wildlife is great for their upbringing ... It is good to see the area maintained as it is part of our lifestyle and why we have chosen to live here." Negative attitudes were illustrated by comments such as: " ... it's a waste of space, always overgrown and I'm scared of snakes living there. It's piled with rubbish. Would love a park for my kids."

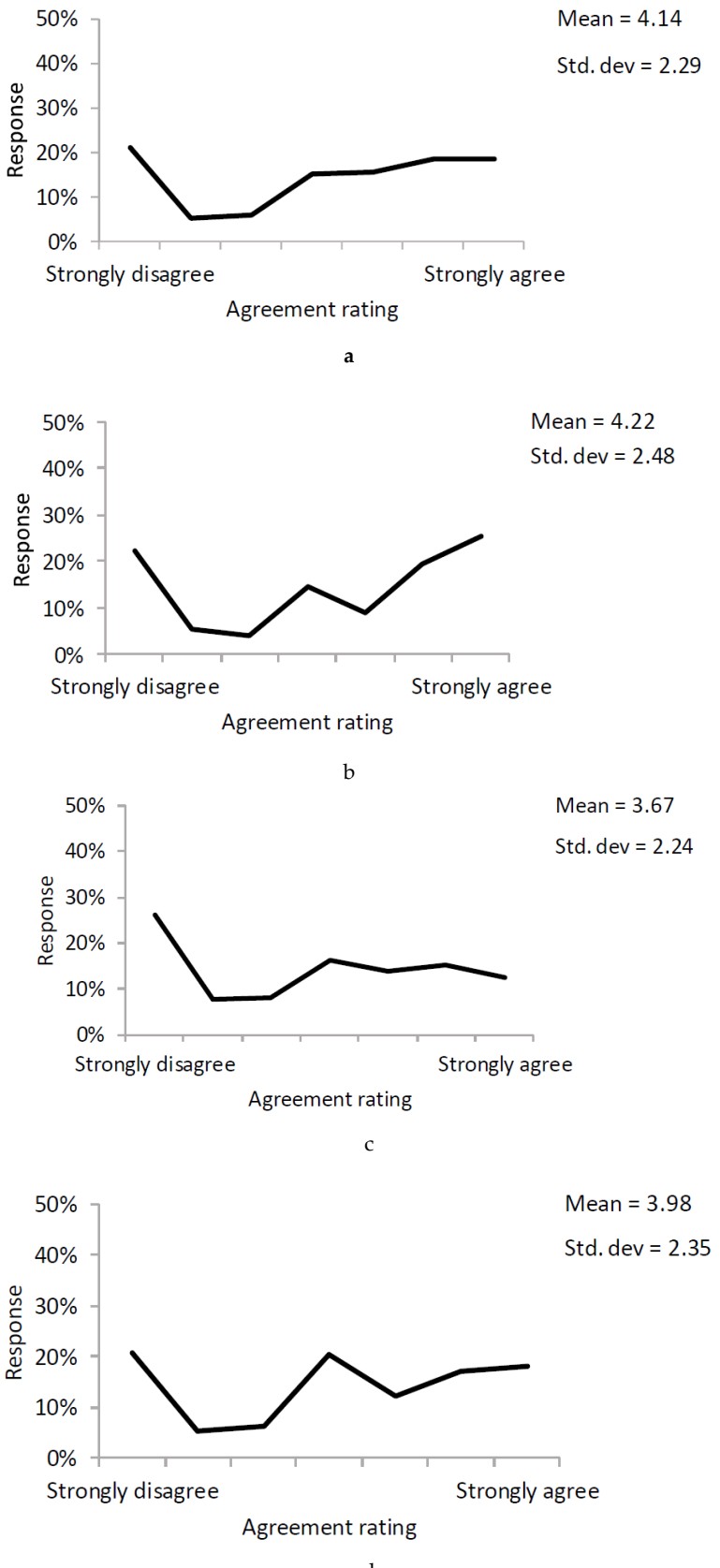

**Figure 2.** Frequency distribution of respondent's (**a**) satisfaction with their local grassland (n = 446), (**b**) enjoyment of using the grassland (n = 440), (**c**) perceptions of how well the local grassland is maintained (n = 444), and (**d**) how safe respondents feel using the local grassland (n = 437).

Nearby residents' self-reported knowledge of the grassland in their neighbourhood was generally high (x = 5.1), (Figure 3). The distribution and some participant comments show there is a small group of residents who have very low self-reported knowledge of their grassland: "we have not been given any information about what [the grassland] is used for, who maintains it or benefits to us as residents and we have been living there for 10 years."

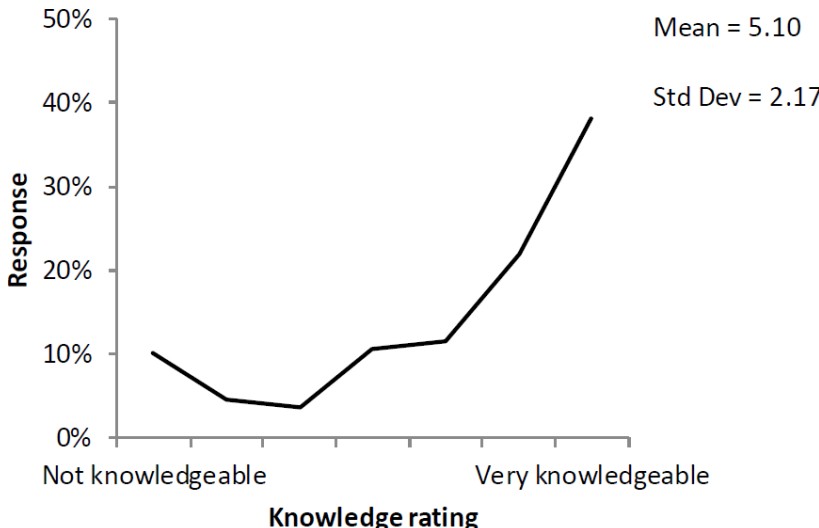

**Figure 3.** Frequency distribution of respondent's self-reported knowledge of the grassland in their neighbourhood (n = 429).

Nearby residents' acceptance of prescribed burning in grasslands in cities was generally high (x = 5.01), although a proportion of respondents found the practice very unacceptable (Figure 4a). This bi-polar distribution indicates there is some conflict between nearby residents in acceptability judgements for prescribed burning. While there is a substantial proportion of public that find prescribed burning acceptable in grasslands in cities, expressed through survey comments: "[the grassland] should be burned once every year there should be far more of it", a group of residents do not find burning acceptable: "it smells. Don't like it. Don't do it near resident housing."

Perhaps unsurprisingly, a substantial proportion of the public found removing trees and shrubs in grasslands unacceptable (x = 3.09); however, the distribution (Figure 4b) and participant comments highlight there is a group of residents who are not sure or perhaps do not feel like they have the knowledge to judge whether this practice is acceptable: "depends whether it replicate[s] the original vegetation cover of the grassland"; "best left to those experts suitably qualified to maintain management practices which keep or return [the] area as close as possible to its original state."

*6.4. Drivers of Attitudes and Acceptance of Management*

Significant relationships between both values and beliefs of individuals (cognitive responses), designed landscape features of grasslands and acceptability and attitude judgments were found (Table 5). Acceptance of prescribed burning was strongly predicted by beliefs that prescribed burning has positive benefits for grasslands (e.g., protects rare or threatened plants and animals living in the grassland), also expressed through participant comments: "[prescribed burning is a] necessary process, good for native flora and fauna"; "it is vital for the survival of the grasslands". There was also a weak relationship between acceptance of prescribed burning and beliefs that prescribed burning has positive benefits for people (e.g., creates a good place for people to meet and socialise), also expressed by survey comments: "feel safer living nearby once burning has occurred", while low acceptance was strongly predicted by beliefs that prescribed burning has negative consequences for people (e.g., is dangerous

for people) expressed through comments such as: "[burning] is dangerous to people and pets"; "smoke from burn[ing] may impact neighbourhood health".

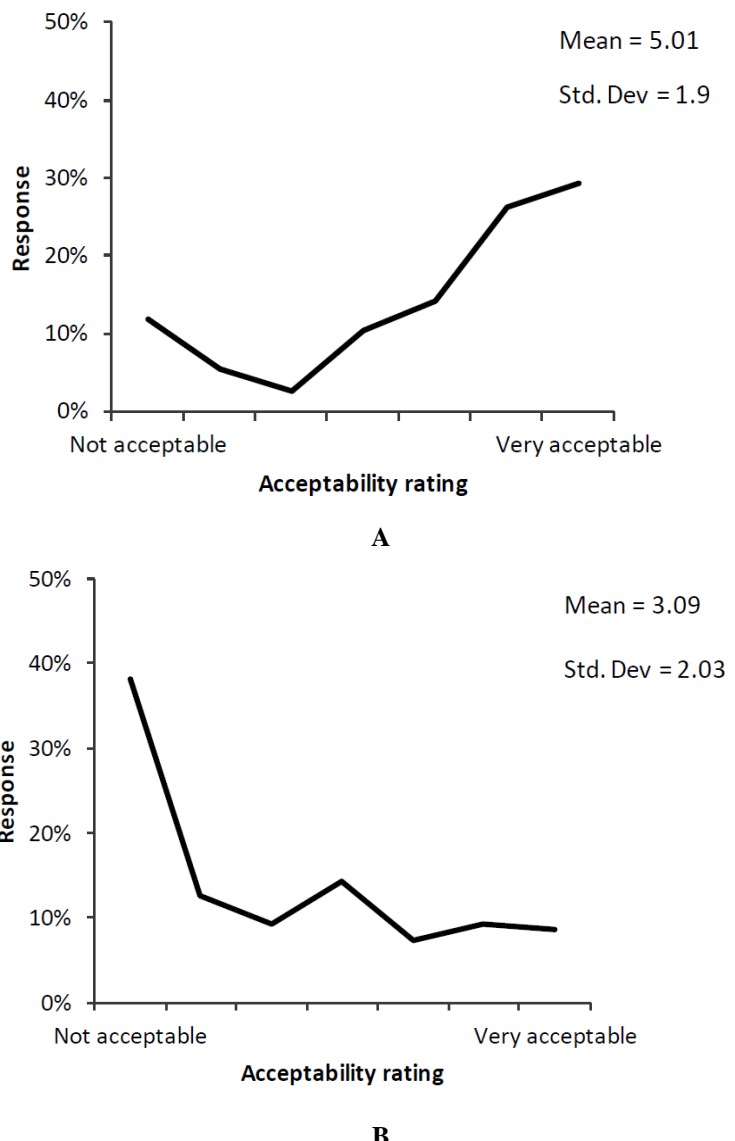

**Figure 4.** Frequency distribution of acceptance ratings for (**a**) using prescribed burning as a management tool (n = 425) and (**b**) removing trees and shrubs from grasslands in cities (n = 418).

Acceptance for removing trees and shrubs from grasslands was predicted by weaker 'natural values' for natural areas in cities expressed by comments: "just get rid of the empty land and put [in] something good, for example a park or houses". Acceptance of removing trees and shrubs was also predicted by beliefs that allowing trees and shrubs to grow in grasslands has negative consequences for people (e.g., is dangerous for people) expressed by comments: "I believe it is acceptable to remove any trees and shrubs if they are at risk of causing harm to people"; "it would worry me if too many trees were to be planted as if a fire was to start I feel it would be more dangerous." There was also a weak relationship between living near a grassland containing interpretive signage and finding removing trees and shrubs from grasslands acceptable.

**Table 5.** Regressions between valued attributes of natural areas in cities, beliefs about consequences of prescribed burning and removing or allowing woody vegetation to grow, self-reported knowledge of the local grassland, designed landscape features of the grasslands and nearby residents attitudes toward the local grassland and acceptance of management actions. Standardised linear regression coefficients are shown *** $p < 0.001$; ** $p < 0.01$; and * $p < 0.05$.

| | Acceptability | | Attitudes | | | |
|---|---|---|---|---|---|---|
| | Acceptance of Planned Burning | Acceptance of Removing Trees and Shrubs from Grasslands | Satisfied with Neighbourhood Grassland | Enjoy Using Neighbourhood Grassland | Feel Safe Using Neighbourhood Grassland | Neighbourhood Grassland is Well Maintained |
| **Valued attributes** | | | | | | |
| Culture and Heritage Values | 0.03 | −0.03 | 0.01 | 0.09 | 0.08 | 0.06 |
| Commercial Use Values | 0.03 | 0.11 * | −0.19 | −0.17 *** | −0.13 ** | −0.16 * |
| Natural Values | 0.03 | −0.21 *** | 0.22 *** | 0.18 *** | 0.15 ** | 0.17 *** |
| Social Values | −0.05 | −0.04 | 0.10 | 0.02 | 0.05 | 0.05 |
| Recreational Setting Values | −0.03 | 0.07 | −0.07 | −0.01 | 0.05 | 0.03 |
| Experiential Values | −0.01 | 0.02 | 0.06 | 0.06 | 0.12 | 0.10 |
| **Beliefs about consequences: Prescribed burning** | | | | | | |
| positive benefits for people | 0.15 ** | | 0.21 *** | 0.26 *** | 0.25 *** | 0.24 *** |
| negative consequences for people | −0.37 *** | | −0.01 | 0.02 | −0.01 | −0.04 |
| positive benefits for grasslands | 0.30 *** | | 0.10 | 0.03 | 0.06 | 0.08 |
| **Beliefs about consequences: Allowing trees and shrubs to grow** | | | | | | |
| positive benefits for people | | 0.02 | −0.04 | 0.10 | 0.03 | 0.00 |
| negative consequences for people | | 0.25 *** | −0.01 | −0.1 | −0.01 | −0.03 |
| positive benefits for grasslands | | 0.06 | 0.09 | 0.01 | 0.06 | 0.07 |
| positive environment for people | | 0.05 | −0.01 | 0.06 | 0.04 | 0.06 |
| **Knowledge** | | | | | | |
| Self-reported knowledge of neighbourhood grassland | 0.06 | 0.07 | 0.20 *** | 0.23 *** | 0.36 *** | 0.26 *** |
| **Designed landscape attributes** | | | | | | |
| Fence | 0.00 | 0.04 | 0.02 | 0.04 | −0.07 | −0.06 |
| Facilities | −0.04 | −0.01 | −0.09 | −0.12 * | −0.02 | −0.11 * |
| Information | 0.01 | 0.12 * | 0.07 | 0.01 | 0.12 ** | 0.13 ** |
| Paths | 0.01 | −0.05 | 0.05 | 0.28 *** | 0.05 | 0.08 |
| $R^2$ | 0.38 | 0.14 | 0.28 | 0.42 | 0.39 | 0.34 |

Positive attitudes toward the grassland people live near can be predicted by strong 'natural values' for natural areas in cities expressed by written comments such as: "please protect [the grassland]. My children and family love the open space and wildlife it attracts like kangaroos, birds etc." Positive attitudes toward neighbourhood grasslands were also predicted by beliefs that prescribed burning has positive benefits for people and positive benefits for grasslands, expressed by the comment: "[I'd] like [prescribed burning] to continue as it takes away a fire hazard and also germinates the plants." Residents with high levels of self-reported knowledge of their local grassland were also more likely to have positive attitudes toward the local reserve. Residents were more likely to feel safe using their local grassland and feel the reserve is being well maintained if they lived near a reserve containing interpretive signage. Residents were also more likely to enjoy using their neighbourhood grassland if they lived near a reserve with a path inside or bordering the reserve also expressed by the comments: "good area for walking and taking in fresh air".

People were less likely to enjoy using their local grassland if they lived near a grassland with notable facilities such as play equipment, barbeque facilities or shelters. This was highlighted by the comments: "concerned when natural grass grows to over 1 metre tall in summertime, fire, snakes, unsafe around child playground"; "[the grassland] was once a cared for, well maintained 'park', where children and families could play, socialise and spend quality time in a safe environment. Now it's classified as 'Native grassland protected' area. Unsightly, unsafe, fire hazard, children unable to play in overgrown [grass]." Negative attitudes toward grasslands were also predicted by strong 'commercial use values' (e.g., utilising the land for shops, cafes and car parks) for natural areas in cities, expressed by the comments: "it should be rezoned for housing and shops. At the moment it feels unsafe and full of snakes"; "allocate the land to build houses so that the council get some money . . . [don't] spend money [on] grass."

## 7. Discussion

We aimed to investigate important knowledge gaps in the management of natural grassland in urban conservation reserves as a socio-ecological system. Our findings show that reintroducing fire promoted native species over exotic species. However, patches of woody plants also supressed the dominant grass, providing a wider range of habitat conditions and could contribute to the persistence of some native species of conservation value, particularly in the absence of fire. We also found (surprisingly) that people living around grasslands found prescribed burning acceptable, but (less surprisingly) the removal of woody vegetation unacceptable. There were conflicted attitudes and acceptability judgements toward grasslands and their management, which were driven both by people's values and beliefs, and designed features of the landscape such as information, paths and facilities.

### 7.1. The Ecological Effects of Fire and Woody Vegetation

There was no difference in native species diversity between recently burnt, unburnt and areas within woody vegetation halos in Melbourne's grasslands. These results support studies that similarly found woody plants do not necessarily result in a decrease in native species diversity [32,33,76] and that woody plant encroachment does not always result in a loss of ecosystem function [33]. These findings demonstrate the benefit of maintaining a diversity of ecosystem processes, if not within sites, then at least between different sites in a grassland ecosystem.

Recent burning did not lead to a significant increase of native species richness, either. This finding suggests much of the native forb seed bank may have been lost during previous prolonged inter-fire intervals. It has been suggested the ongoing survival of grasslands requires remediation efforts [12], with recent studies indicating there is potential for the use of direct sown seed mixes to re-establish native grassland assemblages [77]. There are many grassland restoration projects currently in planning or implementation using this innovative mix of horticultural and ecological principles to achieve the restoration of grasslands across south-eastern Australia [78].

However, the halo areas under woody plants, where grasses and forbs grow in shaded conditions and in direct competition with woody species for moisture and nutrients, supported a different suite of species compared to recently burnt and unburnt areas. This was demonstrated most strikingly through a shift in dominance from high SLA (Specific Leaf Area) grasses with a C4-photosynthetic pathway to low SLA grasses with a C3-photosynthetic pathway. C4 grasses with high SLA values are typically expected to favour resource rich environments, for example, environments with high sunlight exposure and higher nutrient and moisture availability [35]. Hence, *Themeda triandra*, which possesses these traits, was the dominant species of the open grassy volcanic plains of south-eastern Australia at the time of European settlement. In contrast, low SLA C3 grasses, which, although native, tend to gain only relatively minor cover in the presence of *T. triandra* [79], are comparatively tolerant of low light intensity and low soil moisture [35]. The shaded and dry environments observed in the more wooded areas of this study are likely to have created conditions outside the physiological tolerance of *T. triandra*, but within the tolerance range of low SLA C3 grasses, facilitating their proliferation. These results suggest that at low densities, woody plants can provide habitat for shade-tolerant and less competitive species, resulting in greater species heterogeneity at the site and entire landscape level [80,81].

C4 grasses were also less common in unburnt areas. The loss of *T. triandra* has been documented previously in grasslands in Southern Australia in the absence of fire. Under these conditions, the species can undergo canopy 'collapse' through the accumulation of dead thatch smothering living tussocks [5]. This may partly explain the reduced presence of C4 species observed in unburnt areas. Species growing in unburnt areas also possessed large seed mass compared to the other environments. This finding is consistent with previous studies hypothesising large seed mass is an advantage for seeds germinating below litter in long unburnt grasslands [82,83], while the recruitment of small-seeded species is enhanced by disturbance, as small-seeded species tend to have a higher light requirement for germination [84].

These results provide support for the hypothesis that habitat filtering is influencing community composition in Melbourne's highly modified grassland system. The habitat filtering theory predicts that local habitat conditions regulate the composition of plant communities by constraining the functional traits of species that can persist [52,54]. It suggests plant communities assemble as a result of environmental filters (e.g., climate, soil) as the majority of species with traits maladapted to novel habitat conditions (e.g., grasslands undergoing woody plant encroachment) decline and better adapted species increase. This theory predicts convergence in the functional traits of both native and exotic species, with species possessing traits that pre-adapt them to the altered abiotic environment advantaged [52,54].

### 7.2. Attitudes toward Grasslands and Their Management

Public perceptions of grasslands and their management were driven by both the values and beliefs of individuals [41–43] and the designed features of landscapes [48,49]. This information will be useful for grassland managers in understanding why the public arrive at certain acceptability and attitude judgements and help to identify potential points of public conflict, essential knowledge for negotiating mutually acceptable management plans.

### 7.3. Public Attitudes toward Melbourne's Grasslands

This study found conflicted attitudes, with some residents expressing positive attitudes and others negative attitudes. In contrast, numerous studies have found low preferences for native grasslands [39,74]. This research identified that these mixed views have some basis in people's values. Strong commercial use values for natural areas in cities predicted negative attitudes toward grasslands. In contrast, strong natural values for natural areas in cities predicted positive attitudes toward grasslands. These mixed findings may also be explained in part by actions taken by grasslands to educate the public and deliberate design decisions made about the grasslands. Increased self-reported knowledge about the local grassland and beliefs that prescribed burning has positive benefits for

people and positive benefits for grasslands led to positive attitudes toward grasslands. Designed landscape features of Melbourne's grasslands also shaped public attitudes; living near a grassland with a path in or around the reserve promoted positive attitudes toward the grassland.

These findings provide a useful insight for managers who may benefit by addressing these values and beliefs when developing management strategies. Managers may benefit from tailoring community engagement strategies to promote or dispel particular beliefs about grasslands and their management [85]. Providing information that is congruent with the different ways the public value grasslands may also prove useful when promoting positive attitudes toward grasslands as people's values are thought to mediate the way people respond to information [75]. These findings support calls from Williams and Marshall [38] who emphasise the need to design grasslands with visual cues of "good management", including public access, which may send strong messages about the value of the grasslands.

This study further identified that living near a grassland containing notable facilities (such as play equipment, barbeque facilities or shelters) may lead to negative attitudes toward the local grassland. While care should be taken in generalising this finding and further exploration is required to understand this relationship, comments from the surveys suggest these negative attitudes could be based on safety concerns about spending time near long grass, thought to be a fire hazard and to harbour snakes. While good landscape design may help to protect and promote grasslands, this finding suggests that there may be some land uses that are incompatible with grasslands. This emphasises the need for managers to better understand the social context in which Melbourne's grasslands persist.

### 7.4. Acceptability of Prescribed Burning

Although there is some conflict about the use of prescribed burning as a management tool in Melbourne's grasslands, nearby residents mostly find the practice acceptable. While this finding was unexpected, it is consistent with studies from North America that found high levels of public acceptance for prescribed burning in forests [26] and near the wildland–urban interface [27,86]. Beliefs were important predictors of the acceptability of prescribed burning in Melbourne's grasslands. People with beliefs that prescribed burning has negative consequences for people were more likely to find the practice unacceptable, while people with beliefs that prescribed burning has positive benefits for grasslands and positive benefits for people were more likely to find the practice acceptable. This is consistent with several studies from North America that found beliefs important in mediating public attitudes and acceptance of prescribed burning in forests [26,27,87].

It is commonly thought that people's beliefs can be influenced by information from personal contacts, the media and scientific studies [47]. People's beliefs may therefore change over time and can consequently lead to changes in acceptability judgments [47,88]. By understanding the cognitive processes leading to the small but significant portion of residents who have a low acceptance of prescribed burning, managers may be able to address these negative beliefs through relevant education strategies or other community engagement programs. Managers may also be able to improve the acceptability of prescribed burning by communicating the positive benefits of burning for the conservation of native grasslands and for people [50].

### 7.5. Acceptability of Removing Woody Vegetation

A substantial proportion of the public found removing trees and shrubs from grasslands unacceptable. Again, values and beliefs were important mediators of acceptability judgements, consistent with findings reported by Jones et al. [89] who found public support or opposition for urban tree management strategies are determined by environmental values, beliefs and pro-tree attitudes. Our findings lend weight to predictions by Alario [37], who suggests that when restoring native grassland ecosystems in an urban landscape, tree eradication programs may be a point of public conflict and may lead to public outrage when "woody weeds" are removed [37,38]. These public expectations contradict ecological beliefs that woody plants reduce biodiversity and ecosystem

resilience in temperate grasslands [90,91]. Williams and Cary [39] found similar contradictions, reporting that people consider the ecological value of temperate grasslands to be significantly less than that of landscapes with large areas of trees. This low appreciation for treeless ecosystems is consistent with landscape preference theories predicting treeless ecosystems may be held in low regard [39,74,92,93]. There are examples where biodiversity values and aesthetic preferences for a landscape are disconnected. Nassauer's [48] seminal work explains that many 'attractive' landscapes have little ecological value, and many landscapes with high ecological value may be considered 'unattractive'. Learning to recognise habitats and enhancing environmental knowledge can influence people's response to landscapes and influence public support for management [49,94].

Interestingly, acceptance of removing trees and shrubs from Melbourne's grasslands was mediated by the presence of interpretive signage at the respondent's local grassland. While care should be taken in generalising this finding and further exploration is required to understand the relationship between the provision of information and peoples judgments, it does offer support to studies which highlight the need for educational programs to raise awareness of landscapes held in low regard [26,39,95].

One of the most pressing issues in conservation management in urbanised landscapes is the establishment of disturbance regimes that are practical to implement in residential areas without compromising the quality of the remnant patches. Many prominent Australian grassland ecologists have called for restoration efforts to look less toward restoring the historic state of grasslands and rather to develop novel ways of conserving important ecosystem components [11]. Thus, further research into the relative benefits of woody plants in grasslands as opposed to the cost could help to achieve both ecologically sound and socially acceptable management regimes.

## 8. Conclusions

Temperate grasslands confined to cities and towns have been transformed by changes to historic biotic and abiotic habitat conditions attributed to urbanisation [2,96]. It is therefore useful to critically evaluate assumptions about important ecological management actions adopted from rural landscape management: prescribed burning and removing or allowing woody vegetation to grow, under these novel habitat conditions.

Contrary to expectations, this study found patches of woody plants did not reduce native species diversity in grasslands. Rather, shading from woody vegetation supressed the dominant grass (*T. triandra*), providing a wider range of habitat conditions suitable for more shade-tolerant species. This new niche condition is particularly important during prolonged inter-fire intervals common in urban grasslands [2], when *T. triandra* can competitively exclude inter-tussock species. The findings indicate areas under woody plants support a different suite of native species, adding to species heterogeneity both at the patch and landscape scale. Our results showed recent burning did not lead to a significant increase of native grassland flora diversity, indicating that the native seed bank may have been lost during prolonged inter-fire intervals. There is potential for the use of direct sown seed mixes post-fire to re-establish inter-tussock species lost from grasslands [77]. The results of this study suggest native species diversity and community heterogeneity in novel, urbanised grasslands can partly be maintained by burning and assisted by the presence of some patches of woody vegetation.

As grasslands of conservation importance become increasingly surrounded by urban development, the conservation of grasslands will benefit from aligning management with public perceptions and informing the public about grassland conservation. This research provides a way of understanding how the public think about grasslands and their management, offering insights which could help to promote acceptance of ecologically important management practices and facilitate positive attitudes toward these landscapes. An important finding is that a large portion of nearby residents found prescribed burning an acceptable management practice, suggesting that prescribed burning should not be avoided due to concerns about public acceptability. This suggests several avenues of public engagement which may improve public acceptance of this ecologically important practice. Our research also identified avenues to improve attitudes toward grasslands by focusing on people's values and beliefs to inform

management strategies and by identifying designed landscape features such as well-maintained pathways which the public respond to positively. Similar to previous studies investigating the plight of Victoria's grasslands, we emphasise the need for 'good grassland design' to negotiate tensions between human experience and the need to protect an ecologically vulnerable landscape. With careful planning and by understanding the ways the public think about their local grassland, grassland conservation has a promising future in Melbourne and in other cities around the world.

**Supplementary Materials:** The following are available online at http://www.mdpi.com/2071-1050/12/8/3461/s1, Figure S1: Sample Questionnaire: Native Grasslands in Your Neighbourhood.

**Author Contributions:** A.F., D.K. and K.J.H.W. conceived the research and developed the experimental design, methods and survey instrument. A.F. and B.J.Z. collected the data. A.F. analysed the data. A.F. and D.K. wrote the manuscript with revisions by B.J.Z. and K.J.H.W. All authors have read and agreed to the published version of the manuscript.

**Funding:** This work was supported by the Myer Foundation and the Baker Foundation and the APC was funded by Dave Kendal.

**Conflicts of Interest:** The authors declare no conflict of interest.

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
