# Peer review of "Social and Ecological Dimensions of Urban Conservation Grasslands and Their Management through Prescribed Burning and Woody Vegetation Removal"

_sustainability, doi:10.3390/su12083461_

Round 1

Reviewer 1 Report

Conserving natural grassland remnants in urban landscapes is an important challenge of urban biodiversity conservation at a global scale.  This paper uses methods from ecology and the social sciences to analyze (i) the outcome of burning and removing woody species in urban grasslands of Melboure, Australia, in terms of (native) species richness and functional traits, and (ii) to disclose public attitudes towards these measures and their relation to underlying values and beliefs. The results demonstrate opportunities and challenges in temperate grassland management in Australian cities and clearly demonstrate the need of communication strategies in face of conflicting attitudes of adjacent residents.

Congratulations to this paper which is nicely written, well referenced and will become an excellent contribution to the field. It could be accepted as it is, with very minor corrections.

Please check this sentence if an “as” should be added: Factors such drought, timing of fire and species identity are likely to be central in determining grassland resilience to a reintroduced burning regime (Moore et al. 2019).

Please add the year in this sentence: Sites were surveyed from October to December, when most vascular species were in flower to aid identification and detectability

Please indicate how many people had been invited to participate in the survey, in this sentence: The resident who was next to celebrate their birthday, and who is over the age of 18 was invited to fill in and return the questionnaire.

Might it be worthy to considerate in the discussion, that the minimal period of time between “recently burnt” and “unburnt” could be too short? (I am no expert in fire ecology).

Please don’t forget to adopt the journal style for the citations.

Author Response

Thanks for your positive comments on our manuscript. We have revised it in response to all your comments and suggestions.

Reviewer 1:

Conserving natural grassland remnants in urban landscapes is an important challenge of urban biodiversity conservation at a global scale.  This paper uses methods from ecology and the social sciences to analyze (i) the outcome of burning and removing woody species in urban grasslands of Melboure, Australia, in terms of (native) species richness and functional traits, and (ii) to disclose public attitudes towards these measures and their relation to underlying values and beliefs. The results demonstrate opportunities and challenges in temperate grassland management in Australian cities and clearly demonstrate the need of communication strategies in face of conflicting attitudes of adjacent residents.

Congratulations to this paper which is nicely written, well referenced and will become an excellent contribution to the field. It could be accepted as it is, with very minor corrections.

Please check this sentence if an “as” should be added: Factors such drought, timing of fire and species identity are likely to be central in determining grassland resilience to a reintroduced burning regime (Moore et al. 2019).

Response to reviewer: Thanks we have corrected this error.

Please add the year in this sentence: Sites were surveyed from October to December, when most vascular species were in flower to aid identification and detectability

Response to reviewer: We have added the year (2014) to this sentence.

Please indicate how many people had been invited to participate in the survey, in this sentence: The resident who was next to celebrate their birthday, and who is over the age of 18 was invited to fill in and return the questionnaire.

Response to reviewer: the number of invitations (n= 2,832) was added to this sentence.

Might it be worthy to considerate in the discussion, that the minimal period of time between “recently burnt” and “unburnt” could be too short? (I am no expert in fire ecology).

Response to reviewer: We have added the qualifier “noting that the fire frequency in these grasslands was historically very high and fires were likely to occur every 1-3 years (Morgan & Salmon, 2020).”

Please don’t forget to adopt the journal style for the citations.

Response to reviewer: We have reformatted the citations according to the journal style.

Reviewer 2 Report

Social and ecological dimensions of urban conservation grasslands and their management through prescribed burning and woody vegetation removal

Farrar, Kendal, Williams, and Zeeman

This is a brief review. While this is an interesting study about social and ecological dimensions of urban grasslands, the presentation of the research is incomplete and, thus, revisions are needed before a more thorough review of the manuscript can be completed. As is, the research is not repeatable; the methods are incomplete.

--Other:

-Figure and table legends are incomplete; need to heavily refer back to main text to interpret; many tables do not state what bolded values indicate;

-Please fix typos/proofread e.g., section number goes from 1.1 to 1.2; the authors refer to a ‘Figure 23’

-Table one states that their 38 sample landscape, in the methods the authors state “('recently burnt' n = 14) or ii) the site contained a patch of woody vegetation (trees or large shrubs) ('woody plant halos' n = 9).” Which sums to 23, right?

--Does the introduction provide sufficient background and include all relevant references?

No. For example, there is no mention of any questions to be answered about native and exotic species in the introduction, yet this is a major component of the analyses, results, and discussion.

Some additional background on why C4 and C3 is important would be helpful for some readers.

--Is the research design appropriate?

I think so. I am unable to fully evaluate the technical soundness as the methods are, in my opinion, incomplete.

--Are the methods adequately described?

No. Statistical procedures and analyses section/information is incomplete, e.g., procedures not listed in the methods are listed in the results, insufficient detail regarding analyses, lack of citations, etc

--Are the results clearly presented?

No. Please consider revising to make the document easier for the reader. Mostly there are a lot of results given the two main avenues of the paper, it can feel long when reading.

--Are the conclusions supported by the results?

To be determined.

-- Have they been fair in their treatment of previous literature?

Fair, sure. However, I think there are several places where there needs to be more citations as well as supplementing current citations with more recent work.

Please see marked up manuscript for more comments and questions.

Author Response

Thanks for your detailed comments on and suggestions for our manuscript. We have revised it to address all the issues you have raised, which we believe has resulted in a much improved manuscript.  

Reviewer 2:

This is a brief review. While this is an interesting study about social and ecological dimensions of urban grasslands, the presentation of the research is incomplete and, thus, revisions are needed before a more thorough review of the manuscript can be completed. As is, the research is not repeatable; the methods are incomplete.

Response to reviewer: Thanks for your detailed review and comments, we have addressed all comments which have improved the manuscript.

 --Other:

-Figure and table legends are incomplete; need to heavily refer back to main text to interpret; many tables do not state what bolded values indicate;

Response to reviewer: We have clarified figure captions and removed unnecessary bolding.

-Please fix typos/proofread e.g., section number goes from 1.1 to 1.2; the authors refer to a ‘Figure 23’

Response to reviewer: We have corrected these errors.

-Table one states that their 38 sample landscape, in the methods the authors state “('recently burnt' n = 14) or ii) the site contained a patch of woody vegetation (trees or large shrubs) ('woody plant halos' n = 9).” Which sums to 23, right?

Response to reviewer: We have corrected table 1 to refer to 23 sample sites.

--Does the introduction provide sufficient background and include all relevant references?

No. For example, there is no mention of any questions to be answered about native and exotic species in the introduction, yet this is a major component of the analyses, results, and discussion.

Response to reviewer: In Australia protection of ecological communities only extends to native species. We have clarified in the methods that we analyse native and exotic species separately “to better understand the effects of the treatments on the threatened ecological community (only native species are protected)”.

Some additional background on why C4 and C3 is important would be helpful for some readers.

Response to reviewer: We have clarified the importance of C3/C4 photosynthetic pathway in the introduction and methods (particularly the description of data trait collection.

--Is the research design appropriate?

I think so. I am unable to fully evaluate the technical soundness as the methods are, in my opinion, incomplete.

--Are the methods adequately described?

No. Statistical procedures and analyses section/information is incomplete, e.g., procedures not listed in the methods are listed in the results, insufficient detail regarding analyses, lack of citations, etc

Response to reviewer: We have expanded the description of methods and added additional citations as requested.

--Are the results clearly presented?

No. Please consider revising to make the document easier for the reader. Mostly there are a lot of results given the two main avenues of the paper, it can feel long when reading.

Response to reviewer: We hope the revised manuscript is more readable and are happy to revise further as requested..

--Are the conclusions supported by the results?

To be determined.

-- Have they been fair in their treatment of previous literature?

Fair, sure. However, I think there are several places where there needs to be more citations as well as supplementing current citations with more recent work.

Response to reviewer: We have added some additional citations including some more recent references as requested.

Please see marked up manuscript for more comments and questions.

Response to reviewer: Thanks for these detailed comments. We have incorporated all suggestions and comments into this revised manuscript.

Round 2

Reviewer 2 Report

The authors of the “Social and ecological dimensions of urban conservation grasslands and their management through prescribed burning and woody vegetation removal” made substantial revisions to the manuscript. Yet there are some lingering issues.

First, importantly, the authors did not provide a point-by-point response to my previous comments, contained within the PDF I attached. The instructions for authors found at https://www.mdpi.com/authors#General_PeerReview_and_Editorial_Procedure for “Reconsider after Major Revisions: The acceptance of the manuscript would depend on the revisions. The author needs to provide a point by point response or provide a rebuttal if some of the reviewer’s comments cannot be revised. Usually, only one round of major revisions is allowed. Authors will be asked to resubmit the revised paper within ten days and the revised version will be returned to the reviewer for further comments.” Please advise.

The authors stated that they revised the manuscript as follows:

“Response to reviewer: In Australia protection of ecological communities only extends to native species. We have clarified in the methods that we analyse native and exotic species separately “to better understand the effects of the treatments on the threatened ecological community (only native species are protected)”.”

 I cannot find this sentence in the updated manuscript. As such, I still think there needs to be more information about the questions they ask and answer about native and exotic species in the introduction.

Specific comments:

Lines 188-196: “…”

This paragraph doesn’t provide a lot of information. Who are the different landholders? How are grasslands managed differently? Etc.

Line 207: “The selected study sites varied from less than 1 ha to 164 ha in size.”

Did grassland size have an effect of the results? Seems like an important covariate. Please address.

Paragraph starting at Line 228:

What software and software functions did you use for these analyses? Citation?

Line 907: Table 2. “Results of SIMPER analysis: average percent dissimilarity between areas within woody plant halos, unburnt areas and recently burnt areas. These species contributed up to 50% of the average dissimilarity between management treatments. Average frequency is the average importance score value. Items in bold indicate significant differences between importance scores. *Denotes exotic species.”

Why are the authors introducing a new concept, “importance scores”, with the table caption. I am still not clear on what this is. Also, the bolding is confusing to me. So maybe it will be to somebody else too.

Tables 3 and 4 – First use of “pattern matrix loading values” is in the table. What are these? What is VALS?  The authors stated in their rebuttal to me that “Response to reviewer: We have clarified figure captions and removed unnecessary bolding.” This does not appear to be true.

Table 5. I cannot tell what the analysis was from the figure caption. “Standardized regression coefficients” is not sufficient.

Items not addressed from first review  (I do hope the authors reviewed my marked-up document from my first review):

Line 234: Please provide citation for PRIMER. I bet it will make the authors of that software happy.

Line 241: Functional diversity has not been defined

Lines 350-351 “The native graminoid, Themeda triandra, possesses a C4 photosynthetic pathway and has relatively high SLA (20.76 mg/ mm²)”

How much is relatively high? Compared to what?

Line 362: “height diversity”

What does this mean?

Figures 2 – 4:

What are these lines?

Did you just connect the means between ratings?

Are these lines means?

Why not present mean and standard error for each Likert option (acceptability)?

Author Response

Response to reviewers

The authors of the “Social and ecological dimensions of urban conservation grasslands and their management through prescribed burning and woody vegetation removal” made substantial revisions to the manuscript. Yet there are some lingering issues.

First, importantly, the authors did not provide a point-by-point response to my previous comments, contained within the PDF I attached. The instructions for authors found at https://www.mdpi.com/authors#General_PeerReview_and_Editorial_Procedure for “Reconsider after Major Revisions: The acceptance of the manuscript would depend on the revisions. The author needs to provide a point by point response or provide a rebuttal if some of the reviewer’s comments cannot be revised. Usually, only one round of major revisions is allowed. Authors will be asked to resubmit the revised paper within ten days and the revised version will be returned to the reviewer for further comments.” Please advise.

Response to reviewer: Apologies it appears you were not sent the latest version of the manuscript for review. We had incorporated all suggested changes in the marked up PDF into the revised manuscript. We do not rebut any of these points.

The authors stated that they revised the manuscript as follows:

“Response to reviewer: In Australia protection of ecological communities only extends to native species. We have clarified in the methods that we analyse native and exotic species separately “to better understand the effects of the treatments on the threatened ecological community (only native species are protected)”.”

 I cannot find this sentence in the updated manuscript. As such, I still think there needs to be more information about the questions they ask and answer about native and exotic species in the introduction.

Response to reviewer: Apologies it appears you were not sent the latest version of the manuscript for review. The revised sentence should be in this latest submission. We have also expanded the introductory paragraph to state “In Australia, a key focus of grassland conservation is the preservation of endemic species and ecological communities which can be outcompeted by introduced species, particularly in conjunction with altered disturbance regimes.”

Specific comments:

Lines 188-196: “…”

This paragraph doesn’t provide a lot of information. Who are the different landholders? How are grasslands managed differently? Etc.

Response to reviewer: We have expanded this statement with more information on land owners and management by adding “e.g. grazing, rural residential living, roadsides, cemeteries, military barracks)”

Line 207: “The selected study sites varied from less than 1 ha to 164 ha in size.”

Did grassland size have an effect of the results? Seems like an important covariate. Please address.

Response to reviewer: We have added a sentence to clarify, and added a new reference “Previous studies have shown that all reserves including small ones have high levels of endemic species diversity [58]”. The use of quadrats allows comparison of species density in different sized reserves.

Paragraph starting at Line 228:

What software and software functions did you use for these analyses? Citation?

 Response to reviewer: Apologies it appears you were not sent the latest version of the manuscript for review – we had added “using Microsoft Excel macro software developed by Lepš et al.”

Line 907: Table 2. “Results of SIMPER analysis: average percent dissimilarity between areas within woody plant halos, unburnt areas and recently burnt areas. These species contributed up to 50% of the average dissimilarity between management treatments. Average frequency is the average importance score value. Items in bold indicate significant differences between importance scores. *Denotes exotic species.”

 Why are the authors introducing a new concept, “importance scores”, with the table caption. I am still not clear on what this is. Also, the bolding is confusing to me. So maybe it will be to somebody else too.

Response to reviewer: We have removed the reference to importance scores and replaced the bold values with symbols to indicate significance

Tables 3 and 4 – First use of “pattern matrix loading values” is in the table. What are these? What is VALS?  The authors stated in their rebuttal to me that “Response to reviewer: We have clarified figure captions and removed unnecessary bolding.” This does not appear to be true.

Response to reviewer: We have removed reference to pattern matrix. The VALS is a published psychometric scale that has been described in the methods. We are happy to provide more information on this if needed. Apologies the bold values had been removed but the incorrect manuscript was provided to you for review.

Table 5. I cannot tell what the analysis was from the figure caption. “Standardized regression coefficients” is not sufficient.

Response to reviewer: We describe in the methods that standard linear regressions were performed using SPSS, and we report standardised regression coefficients in the table. We are unclear what additional information is required but happy to add extra information if needed.

Items not addressed from first review ☹ (I do hope the authors reviewed my marked-up document from my first review):

Response to reviewer: Apologies it appears you were not sent the latest version of the manuscript for review. We had incorporated all suggested changes in the marked up PDF into the revised manuscript. We do not rebut any of these points.

Line 234: Please provide citation for PRIMER. I bet it will make the authors of that software happy.

 Response to reviewer: We had included a citation for PRIMER.

Line 241: Functional diversity has not been defined

 Response to reviewer: We had added a clarification of what we mean by functional diversity “(the diversity of functional traits)”

Lines 350-351 “The native graminoid, Themeda triandra, possesses a C4 photosynthetic pathway and has relatively high SLA (20.76 mg/ mm²)”

How much is relatively high? Compared to what?

 Response to reviewer: We had added the clarification “compared with C3 graminoids”

Line 362: “height diversity”

What does this mean?

Response to reviewer: We had clarified the text to say that : “as measured by the Rao index of trait diversity. This difference is driven by exotic species, with the diversity of plant height in exotic species…”

Figures 2 – 4:

What are these lines?

Did you just connect the means between ratings?

Are these lines means?

Why not present mean and standard error for each Likert option (acceptability)?

Response to reviewer: These graphs are frequency graphs showing the distribution of responses. The y-axis shows proportion of respondents, so does not show a mean or have an error bar.